# Input Dimension Expandable Network: Integrating New Input Dimensions in Online Learning

## Abstract

Sensory augmentation experiments have demonstrated that the perceptual dimensions of the mammalian nervous system are expandable at different levels. This capacity enables mammals to acquire signals beyond the range of their inherent sensory systems and subsequently learn to utilize such signals. A critical question arises: **how to enable a learning system to expand its input dimensions in an online manner?** To address this challenge, we propose a hierarchical modular neural network architecture that supports multi-level and multi-regional expansion of input dimensions, along with a dimension integration algorithm designed to guide new dimensions to proper neuron circuits during online learning. To validate our computational model, we design a series of dimension expansion experiments at different levels. The experimental results confirm that our method effectively handles the input dimension expandable learning problem.

## 1 Introduction

As shown in Fig. 1(a), neuroscientists have explored sensory augmentation through brain-computer interface Thomson et al. (2017) and genetic engineering Zhang et al. (2017). For example, researchers have successfully enabled rats to perceive infrared light Thomson et al. (2017) via a brain-computer interface; made the red color-blind mice Zhang et al. (2017) acquire red perception ability by the gene therapy that injects human opsin gene into their retina.

Notably, these signals are entirely imperceptible to naturally occurring animals! Furthermore, the animals were able to learn to utilize their newly acquired perceptual capacities to perform specific tasks, such as color discrimination and signal localization, which are otherwise impossible for ordinary animals. Collectively, these studies lead to a definitive conclusion: the perceptual dimensions of the mammalian nervous system can be expanded in a plug-and-play manner, and mammals can learn to leverage these expanded dimensions.

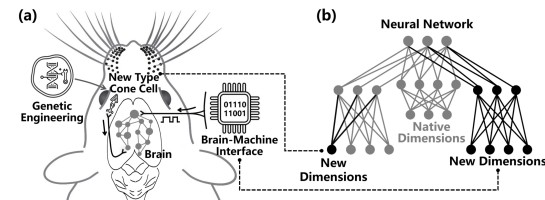

Figure 1: (a) The expansion of perception dimensions at different levels of the mammalian nervous system, e.g., at the retina and at the cerebral cortex. (b) Schematic diagram of a neural network model that can expand input dimensions. New input dimensions (represented by the black circles) can be introduced at different levels of the neural network.

The above conclusion enlightens a computational problem: **how to introduce the expansibility of perceptual dimensions into a learning system?** If the problem is solved, the learning system will be able to expand its input dimensions during its lifetime, as shown in Fig. 1(b). Such a system can be applied to a variety of fields including robot system, data stream mining, and information fusion. For example, if we want to install a microphone to a robot which uses only a camera as an input device, with the system, there is no need to retrain the robot offline from scratch; the information gathered from the microphone is integrated with the information gathered from the camera online automatically!

In recent years, many researchers have studied the problem from different perspectives. However, to the best of our knowledge, no existing method is capable of simultaneously supporting the online expansion of input dimensions across both multiple levels and multiple regions. Our method is the first approach that enables such online expansion of input dimensions at multiple levels and regions. Briefly, the main contributions of our work include: (1) We propose a hierarchical modular neural network capable of expanding input dimensions across all hierarchical levels and modular areas. The input dimensions of the network can be expanded at any time during online learning. (2) We introduce a dimension integration algorithm that integrates information from new dimensions with the neural circuits of existing dimensions, which enables the network to absorb the novel perceptual dimensions in an online way. (3) We design corresponding experiments to realize and verify the dimension expandable learning paradigm.

## 2 RELATED WORK

Relevant fields include machine learning with incremental features Zhou (2022) and machine learning in open feature space Xing et al. (2021); He et al. (2023). Here, we briefly review these fields.

Some studies directly expand the input dimension of the learning machine for the new features. Xing et al. (2016); Peng et al. (2021) learn and store features of new dimensional data by expanding the dimension of cluster center vectors learned in the original dimensional space, thereby integrating new dimensional features into cluster center vectors. Hou et al. (2023); You et al. (2024) expand the input dimension of a linear classifier to receive new dimensional features, then use full-dimensional (original and new) features to train the classifier. However, after training with the new input dimensions, these methods are unable to perform classification on the data from the original dimensions.

Some studies hypothesize that there is a relationship between the new dimensional features and the original dimensional features. Hou et al. (2019; 2022) assume that there is a linear mapping between the new dimensional data and the original dimensional data. They learn the mapping by reconstructing the old dimensional data with the new dimensional data, then use the mapping to classify new dimensional data. Subsequently, Hou et al. (2021); Lian et al. (2024) extend the linear mapping to a nonlinear mapping. Wang & Mo (2021) learn the relationship by cross-feature attention layers and obtain parameters of each layer by minimizing the within-class scattering and maximizing the interclass scattering. Similarly, these methods are also unable to perform classification using data from the original dimensions after new dimensions have been introduced.

Some studies integrate the new input dimensions by ensemble learning. Hou & Zhou (2018); Liu et al. (2021; 2022); Tu et al. (2025) train two classifiers in the original and new dimensional spaces respectively, and integrate the two classifiers using ensemble learning. Gu et al. (2024) train a set of classifiers in both existing and new dimensional spaces and then integrate them. Schreckenberger et al. (2023) train a feature forest for each new dimension and integrate these feature forests with the forests trained on existing dimensions. However, after training with the new input dimensions, these methods must use data from all dimensions for classification and cannot use data from the original dimensions or new dimensions independently.

## 3 INPUT DIMENSION EXPANDABLE NETWORK

### 3.1 NETWORK ARCHITECTURE

As mentioned before, the input dimension of the mammalian nervous system can be expanded at multiple levels. To enable the expansion of input dimensions at multiple levels and across multiple areas, a hierarchical and modular neural network is designed. As shown in Fig. 2, we adopt the structure of the brain network. It is a multi-channel structure, where each channel consists of several feature areas and a unimodal association layer. A multimodal association layer is positioned on top to integrate all channels. Signals are transmitted in both ascending and descending directions between different channels, which enables the mutual activation of neurons from different channels (modalities) and thus facilitates cross-modal recall. The expansion of input dimensions can occur at all layers and areas. We use $N^{\alpha_k}$ to represent the set of feature neurons of type $\alpha_k$, $N^\beta$ to represent the set of unimodal association neurons in channel $\beta$, and $N$ to represent the set of multimodal association neurons.

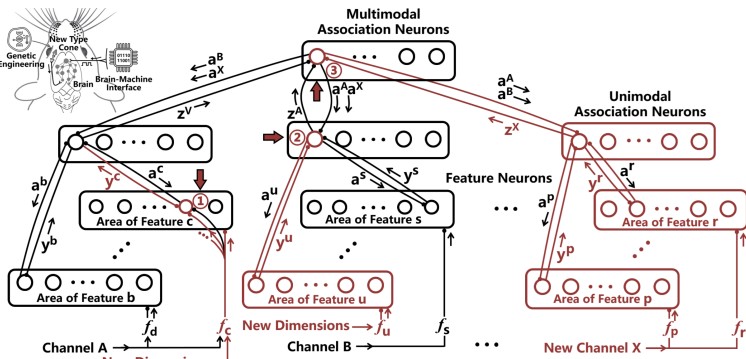

Figure 2: Input dimension expandable network. Signals are transmitted in both ascending and descending directions. Dimension expansion can occur at all layers, as marked by the circled numbers.

### 3.1.1 FEATURE NEURON

As shown in Fig. 2, a feature neuron responds to some particular type of features. We use $N^{\alpha_k}$ to represent the set of feature neurons of type $\alpha_k$. Each neuron $N_j^{\alpha_k}$ has an ascending pathway and a descending pathway. The ascending pathway receives feature vector $\boldsymbol{x} = [x_1, x_2, ..., x_n]$. The ascending activation function $f_F^a$ of $N_j^{\alpha_k}$ is defined as follows,

$$f_F^a = \begin{cases} \boldsymbol{y}^{\alpha_k} = \sum_{i=1}^{n} \sum_{t=1}^{T} w_{j,i} \cos \lambda_i^{\alpha_k} 2\pi \frac{t-1}{T}, & \|\boldsymbol{x} - \boldsymbol{w}_j\|_2 \leq \theta \\ \\ 0, & \text{otherwise} \end{cases} \tag{1}$$

where $\boldsymbol{w}_j = [w_{j,1}, w_{j,2}, ..., w_{j,n}]$ and $\theta$ are the weights and threshold of $N_j^{\alpha_k}$. $\boldsymbol{y}^{\alpha_k}$ is an activation signal which will be transmitted to the unimodal association neurons to which $N_j^{\alpha_k}$ connects. $\lambda_i^{\alpha_k}$ is a frequency parameter which corresponds to the $i$-th dimension of the weights (or features) with type $\alpha_k$. Here, each dimension corresponds to a unique frequency which means each feature type $\alpha_k$ corresponds to a unique frequency vector $\boldsymbol{\lambda}^{\alpha_k}$ in the network. We assign a unique natural number to each $\lambda_i^{\alpha_k}$ in practice. $T$ is a parameter which is used to generate a period time of signal.

The descending pathways receive signals from unimodal association neurons. $\boldsymbol{U}_{i,j}^{\alpha_k} = 1$ means there exists a descending connection from unimodal association neuron $N_i^{\beta}$ to feature neuron $N_j^{\alpha_k}$ and $\boldsymbol{U}_{i,j}^{\alpha_k} = 0$ means not. We use $\boldsymbol{a}^{\alpha_k} = [a_1^{\alpha_k}, a_2^{\alpha_k}, ..., a_m^{\alpha_k}]$ to represent a signal transmitted in a descending pathway between a unimodal association neuron and a feature neuron, and $\boldsymbol{A}^{\alpha_k} = [A_1^{\alpha_k}, A_2^{\alpha_k}, ..., A_m^{\alpha_k}]$ to denote the signal variable. Each dimension $A_i^{\alpha_k}$ corresponds to a frequency $\lambda$, which means this dimension receives an amplitude value $a_i^{\alpha_k}$ at frequency $\lambda$. $A_i^{\alpha_k}$ is modeled as a Gaussian distribution $A_i^{\alpha_k} \sim \boldsymbol{N}(\mu_i, \sigma_i)$, and a relative probability density of a sample $a_i^{\alpha_k}$ of $A_i^{\alpha_k}$ is calculated with

$$p_i^{\alpha_k} = \exp(-\frac{(a_i^{\alpha_k} - \mu_i)^2}{2\sigma_i^2}), \quad 1 \leq i \leq m$$

The descending activation function in this descending pathway is defined as follows,

$$f_F^d = \begin{cases} 1, & \forall p_i^{\alpha_k} \geq \vartheta, \ 1 \leq i \leq m \\ 0, & \text{otherwise} \end{cases} \tag{2}$$

where $\vartheta$ is the threshold for the relative probability density.

### 3.1.2 UNIMODAL ASSOCIATION NEURON

As shown in Fig. 2, a unimodal association neuron associates different types of feature neurons in a channel to form a unimodal concept. For example, the unimodal association neurons in a visual channel can associate shape, color and other types of feature neurons to form visual concepts. We use $N^{\beta}$ to represent the set of unimodal association neurons in channel $\beta$.

The ascending connections from feature neurons of type $\alpha_k$ to unimodal association neurons are represented by a 0-1 matrix $\boldsymbol{W}^{\alpha_k}$, where $\boldsymbol{W}_{i,j}^{\alpha_k} = 1$ means there exists a connection from feature

neuron $N_j^{\alpha_k}$ to unimodal association neuron $N_i^{\beta}$ and $\boldsymbol{W}_{i,j}^{\alpha_k} = 0$ means not. Assume that there are $n$ different feature areas $\alpha_1, \alpha_2, ..., \alpha_n$ in channel $\beta$, the ascending activation function of the unimodal association neuron $N_i^{\beta}$ is defined as follows,

$$f_U^a = \begin{cases} \boldsymbol{z}^{\beta} = \sum_{k=1}^{n} \boldsymbol{y}^{\alpha_k}, & \forall \boldsymbol{W}_{i,:}^{\alpha_k} \cdot \boldsymbol{e}^{\alpha_k} = 1, \ 1 \leq k \leq n \\ \\ 0, & \text{otherwise} \end{cases} \tag{3}$$

where $\boldsymbol{W}_{i,:}^{\alpha_k}$ is the $i$-th row of $\boldsymbol{W}^{\alpha_k}$, $\boldsymbol{e}^{\alpha_k}$ is a 0-1 vector, $e_j^{\alpha_k} = 1$ if feature neuron $N_j^{\alpha_k}$ is activated. $\boldsymbol{y}^{\alpha_k}$ represents the signals generated by an activated feature neuron with feature type $\alpha_k$ in channel $\beta$ using Eq. (1). $\boldsymbol{z}^{\beta}$ is the activation signal of $N_i^{\beta}$ which equals the sum of the signals of feature neurons to which $N_i^{\beta}$ connects.

The descending pathways receive signals from multimodal association neurons. $\boldsymbol{U}_{j,i}^{\beta} = 1$ means there exists a descending connection from multimodal association neuron $N_j$ to unimodal association neuron $N_i^{\beta}$ and $\boldsymbol{U}_{j,i}^{\beta} = 0$ means not. We use $\boldsymbol{a}^{\beta} = [a_1^{\beta}, a_2^{\beta}, ..., a_s^{\beta}]$ to represent a signal transmitted in a descending pathway between a multimodal association neuron and a unimodal association neuron, and $\boldsymbol{A}^{\beta} = [A_1^{\beta}, A_2^{\beta}, ..., A_s^{\beta}]$ to denote the signal variable. Each dimension $A_i^{\beta}$ corresponds to a frequency $\lambda$, which means this dimension receives an amplitude value $a_i^{\beta}$ at frequency $\lambda$. The descending activation function is modeled similarly to Eq. (2),

$$f_U^d = \begin{cases} \boldsymbol{a}^{\alpha_k}, & \forall p_i^{\beta} \geq \vartheta, \ 1 \leq i \leq s \\ 0, & \text{otherwise} \end{cases} \tag{4}$$

where $\boldsymbol{a}^{\alpha_k}$ is a descending signal which is transmitted to feature area $\alpha_k$ as shown in Fig. 2.

### 3.1.3 MULTIMODAL ASSOCIATION NEURON

As shown in Fig. 2, multimodal association neurons connect unimodal association neurons across channels. They transmit signals from one channel to others, enabling coordination among channels. We denote the set of multimodal association neurons as $N$.

Ascending connections from unimodal association neurons in channel $\beta$ to multimodal association neurons are represented by a 0-1 matrix $\boldsymbol{W}^{\beta}$, where $\boldsymbol{W}_{i,j}^{\beta} = 1$ means there exists an ascending connection from unimodal association neuron $N_j^{\beta}$ to multimodal association neuron $N_i$ and $\boldsymbol{W}_{i,j}^{\beta} = 0$ means not. The ascending activation function of $N_i$ is defined as follows,

$$f_M^a = \begin{cases} [\boldsymbol{a}, \boldsymbol{\lambda}] = \mathcal{F}(\boldsymbol{z}^{\beta}), & \boldsymbol{W}_{i,:}^{\beta} \cdot \boldsymbol{e}^{\beta} = 1 \\ 0, & \text{otherwise} \end{cases} \tag{5}$$

where $\boldsymbol{W}_{i,:}^{\beta}$ is the $i$-th row of $\boldsymbol{W}^{\beta}$. $\boldsymbol{e}^{\beta}$ is a 0-1 vector, $e_j^{\beta} = 1$ when unimodal association neuron $N_j^{\beta}$ is activated. $\boldsymbol{z}^{\beta}$ is the output of the activated unimodal association neuron in channel $\beta$ generated by Eq. (3), $\mathcal{F}()$ is the Fourier transform. The output $[\boldsymbol{a}, \boldsymbol{\lambda}]$ are the amplitude and frequency obtained by the Fourier transform, and we write $\boldsymbol{a}^{\beta} = [\boldsymbol{a}, \boldsymbol{\lambda}]$. The amplitude $\boldsymbol{a}$ can be transmitted to the unimodal association layer and feature areas of all other channels via descending connections according to the signal variable $\boldsymbol{A}^{\beta}$ and frequency $\boldsymbol{\lambda}$ attached to $\boldsymbol{A}^{\beta}$ of the descending connections.

### 3.2 DIMENSION INTEGRATION

When new input dimensions are introduced, the neurons should absorb the signals transmitted by the new input dimensions. Now, we introduce the dimension integration of different neurons.

### 3.2.1 FEATURE NEURON

Assume that there are $r$ feature neurons in feature area $\alpha_k$ which receive $n$ dimensional input feature originally. Now new $m$ dimension feature input are introduced, as indicated by the circled number 1 in Fig. 2. The input feature $\bar{\boldsymbol{x}} = [x_1, x_2, ..., x_n] \in \boldsymbol{R}^n$ evolves to $\boldsymbol{x} = [\bar{\boldsymbol{x}}, \tilde{\boldsymbol{x}}] = [x_1, x_2, ..., x_n, ..., x_{n+1}, ..., x_{n+m}] \in \boldsymbol{R}^{n+m}$. During the dimension integration, area $\alpha_k$ can include two types of feature neurons, one type that has already absorbed new input dimensions and the other that has not. We use set $\bar{N}^{\alpha_k}$ and $\tilde{N}^{\alpha_k}$ to represent the sets of neurons whose weights are in $\boldsymbol{R}^n$ and

$\boldsymbol{R}^{n+m}$, $N^{\alpha k} = \bar{N}^{\alpha k} \cup \tilde{N}^{\alpha k}$. The main idea of the integration algorithm is to find an $n$ dimensional feature neuron to absorb the current input $\tilde{\boldsymbol{x}} = [x_{n+1}, x_{n+2}, ..., x_{n+m}]$. First, the network finds feature neurons whose weights are most similar to the current input $\boldsymbol{x}$ in space $\boldsymbol{R}^n$ and $\boldsymbol{R}^{n+m}$,

$$N_{\bar{a}}^{\alpha k} = \underset{N_j^{\alpha k} \in N^{\alpha k}}{\arg\min} \|\bar{\boldsymbol{x}} - \bar{\boldsymbol{w}}_j\|_2, \qquad N_a^{\alpha k} = \underset{N_j^{\alpha k} \in \tilde{N}_a^{\alpha k}}{\arg\min} \|\boldsymbol{x} - \boldsymbol{w}_j\|_2$$

where $\bar{\boldsymbol{w}}_j = [w_{j,1}, w_{j,2}, ..., w_{j,n}]$ is the part of the weights of $N_j^{\alpha k}$ in space $\boldsymbol{R}^n$. Then, the activation function is checked for $N_{\bar{a}}^{\alpha k}$ and $N_a^{\alpha k}$ with Eq. (1). Obviously, there are four combinations:

(1) $\|\bar{\boldsymbol{x}} - \bar{\boldsymbol{w}}_{\bar{a}}\|_2 \le \theta$ and $\|\boldsymbol{x} - \boldsymbol{w}_a\|_2 > \theta$, which means $N_{\bar{a}}^{\alpha k}$ is activated but $N_a^{\alpha k}$ is not.

If the dimension of the weights $\boldsymbol{w}_{\bar{a}}$ of $N_{\bar{a}}^{\alpha k}$ is equal to $n$, the current input activates a low-dimensional feature neuron but does not activate any high-dimensional feature neurons. The network updates and expands the weights $\boldsymbol{w}_{\bar{a}}$ of $N_{\bar{a}}^{\alpha k}$ to a high dimensional space as follows,

$$\boldsymbol{w}_{\bar{a}} = \boldsymbol{w}_{\bar{a}} + \frac{1}{\boldsymbol{u}_{\bar{a}}} \circ (\bar{\boldsymbol{x}} - \boldsymbol{w}_{\bar{a}})$$

$$\boldsymbol{w}_{\bar{a}} = [\boldsymbol{w}_{\bar{a}}, \tilde{\boldsymbol{x}}] \tag{6}$$

where $\boldsymbol{u}_{\bar{a}}$ is the number of updating times of each dimension of $\bar{\boldsymbol{w}}_{\bar{a}}$, $\circ$ is the Hadamard product. Eq. (6) expands the weights of $N_{\bar{a}}^{\alpha k}$ to space $\boldsymbol{R}^{n+m}$ with $\tilde{\boldsymbol{x}}$, which is used to initialize the weights of the new input dimensions. Meanwhile, the ascending activation function of $N_{\bar{a}}^{\alpha k}$ evolves from an $n$ dimension function to an $n + m$ dimension function, i.e.,

$$\|\bar{\boldsymbol{x}} - \bar{\boldsymbol{w}}_{\bar{a}}\|_2 \le \theta \rightsquigarrow \|\boldsymbol{x} - \boldsymbol{w}_{\bar{a}}\|_2 \le \theta \tag{7}$$

$N_{\bar{a}}^{\alpha k}$ activates and generates ascending signals using $n$ original and $m$ new dimensions by Eq. (1),

$$\boldsymbol{y}^{\alpha k} = \sum_{i=1}^{n+m} \sum_{t=1}^{T} w_{\bar{a},i} \cos \lambda_i^{\alpha k} 2\pi \frac{t-1}{T} \tag{8}$$

where $w_{\bar{a},i}$ is the $i$-th element of $\boldsymbol{w}_{\bar{a}}$. $\tilde{\boldsymbol{\lambda}}^{\alpha k} = [\lambda_{n+1}^{\alpha k}, \lambda_{n+2}^{\alpha k}, ..., \lambda_{n+m}^{\alpha k}]$ is initialized and allocated to area $\alpha_k$. In practice, we set them to different natural numbers which are different from all the existing frequency parameters in the network.

If the dimension of $N_{\bar{a}}^{\alpha k}$ is equal to $n+m$, $N_{\bar{a}}^{\alpha k}$ does not need to be expanded to the high-dimensional space $\boldsymbol{R}^{n+m}$. The network creates a new neuron $N_{r+1}^{\alpha k}$ to record the current feature $\boldsymbol{x}$, i.e., $\boldsymbol{w}_{r+1} = \boldsymbol{x}$. Then $N_{r+1}^{\alpha k}$ is activated and generates signals with Eq. (1).

(2) $\|\bar{\boldsymbol{x}} - \bar{\boldsymbol{w}}_{\bar{a}}\|_2 \le \theta$ and $\|\boldsymbol{x} - \boldsymbol{w}_a\|_2 \le \theta$, which means $N_{\bar{a}}^{\alpha k}$ and $N_a^{\alpha k}$ are both activated. In such a scenario, we let the activated high-dimensional neuron $N_a^{\alpha k}$ inhibit the activated low-dimensional neuron $N_{\bar{a}}^{\alpha k}$ and update the weights of neuron $N_a^{\alpha k}$ with $\boldsymbol{w}_a = \boldsymbol{w}_a + \frac{1}{\boldsymbol{u}_a} \circ (\boldsymbol{x} - \boldsymbol{w}_a)$. Then $N_a^{\alpha k}$ is activated and generates signals with Eq. (1).

(3) $\|\bar{\boldsymbol{x}} - \bar{\boldsymbol{w}}_{\bar{a}}\|_2 > \theta$ and $\|\boldsymbol{x} - \boldsymbol{w}_a\|_2 \le \theta$, i.e., $N_a^{\alpha k}$ is activated but $N_{\bar{a}}^{\alpha k}$ is not. The network updates $N_a^{\alpha k}$ and generates ascending signals as the way in combination (2).

(4) $\|\bar{\boldsymbol{x}} - \bar{\boldsymbol{w}}_{\bar{a}}\|_2 > \theta$ and $\|\boldsymbol{x} - \boldsymbol{w}_a\|_2 > \theta$, which means neither $N_{\bar{a}}^{\alpha k}$ nor $N_a^{\alpha k}$ is activated. A new neuron $N_{r+1}^{\alpha k}$ is created and activated as the way in combination (1).

### 3.2.2 Unimodal Association Neuron

When new input dimensions form a new type of feature, unimodal association neurons absorb the new dimensions as the circled number 2 in Fig. 2 shows. For example, channel $B$ has a feature area $s$ originally. Next, a new feature extraction function $f_u$ emerges in the channel. Correspondingly, the input dimension of channel $B$ is expanded by a new feature area $u$.

Assume that there are $r$ unimodal association neurons in channel $\beta$ which receive input from $n$ feature areas originally. Then $m$ new feature areas are introduced. Now, a group of feature neurons $N_{a_1}^{\alpha_1}, N_{a_2}^{\alpha_2}, ..., N_{a_n}^{\alpha_n}$ are activate in the original feature areas $\alpha_1, \alpha_2, ..., \alpha_n$, another group of feature neurons $N_{a_{n+1}}^{\alpha_{n+1}}, N_{a_{n+2}}^{\alpha_{n+2}}, ..., N_{a_{n+m}}^{\alpha_{n+m}}$ are activated in the new feature areas $\alpha_{n+1}, \alpha_{n+2}, ..., \alpha_{n+m}$. Similar to the feature neuron, during the dimension integration, channel $\beta$ can include two types of unimodal association neurons, one type that has already absorbed new input dimensions which are

recorded in set $\tilde{N}^\beta$ and the other that has not which are recorded in set $\bar{N}^\beta$, and $N^\beta = \bar{N}^\beta \cup \tilde{N}^\beta$. We let the activated feature neurons try to activate some unimodal association neuron in sets $\bar{N}^\beta$ and $\tilde{N}^\beta$ with Eq. (3),

$$N_{\bar{a}}^\beta = \{N_i^\beta | \boldsymbol{W}_{i,:}^{\alpha_k} \cdot \boldsymbol{e}^{\alpha_k} = 1,\ 1 \le k \le n,\ N_i^\beta \in \bar{N}^\beta\}$$

$$N_a^\beta = \{N_i^\beta | \boldsymbol{W}_{i,:}^{\alpha_k} \cdot \boldsymbol{e}^{\alpha_k} = 1,\ 1 \le k \le n+m,\ N_i^\beta \in \tilde{N}^\beta\}$$

where $\boldsymbol{e}^{\alpha_k}$ is a 0-1 vector and $\boldsymbol{e}_{a_k}^{\alpha_k} = 1$ which means $N_{a_k}^{\alpha_k}$ is activated. The integration algorithm finds a unimodal association neuron to associate activated feature neurons in the new feature areas according to the results of $N_{\bar{a}}^\beta$ and $N_a^\beta$. Obviously, there are four combinations:

(1) $N_{\bar{a}}^\beta \ne \varnothing$ and $N_a^\beta = \varnothing$, the current input activates a low-dimensional unimodal association neuron but does not activate any high-dimensional unimodal association neurons. The network applies the dimension integration to $N_{\bar{a}}^\beta$ as follows,

$$\boldsymbol{W}_{\bar{a},:}^{\alpha_i} = \boldsymbol{e}^{\alpha_i}, \quad n+1 \le i \le n+m \tag{9}$$

Eq. (9) expands the input dimensions of $N_{\bar{a}}^\beta$ by creating new ascending connections from feature neuron $N_{a_{n+1}}^{\alpha_{n+1}}, N_{a_{n+2}}^{\alpha_{n+2}}, ..., N_{a_{n+m}}^{\alpha_{n+m}}$ in each new feature area to $N_{\bar{a}}^\beta$. Then it generates signals using $n$ original and $m$ new dimensions by Eq. (3),

$$\boldsymbol{z}^\beta = \sum_{i=1}^{n} \boldsymbol{y}^{\alpha_i} + \sum_{i=n+1}^{n+m} \boldsymbol{y}^{\alpha_i} \tag{10}$$

The ascending activation function of $N_{\bar{a}}^\beta$ evolves from $n$ dimension to $n+m$ dimension,

$$\forall \boldsymbol{W}_{\bar{a},:}^{\alpha_k} \cdot \boldsymbol{e}^{\alpha_k} = 1,\ 1 \le k \le n \quad \rightsquigarrow \quad \forall \boldsymbol{W}_{\bar{a},:}^{\alpha_k} \cdot \boldsymbol{e}^{\alpha_k} = 1,\ 1 \le k \le n+m \tag{11}$$

(2) $N_{\bar{a}}^\beta \ne \varnothing$ and $N_a^\beta \ne \varnothing$, the current input simultaneously activates a low-dimensional and a high-dimensional unimodal association neuron. The activated high-dimensional neuron $N_a^\beta$ inhibits the activated low-dimensional neuron $N_{\bar{a}}^\beta$ and generates ascending signals with Eq. (10).

(3) $N_{\bar{a}}^\beta = \varnothing$ and $N_a^\beta \ne \varnothing$, the current input activates a high-dimensional unimodal association neuron but does not activate any low-dimensional unimodal association neurons. The activated high-dimensional neuron generates ascending signals with Eq. (10).

(4) $N_{\bar{a}}^\beta = \varnothing$ and $N_a^\beta = \varnothing$, the current input does not activate any unimodal association neurons. A new unimodal association neuron $N_{r+1}^\beta$ is created to associate the activated feature neurons to form a unimodal concept,

$$\boldsymbol{W}_{r+1,:}^{\alpha_k} = \boldsymbol{e}^{\alpha_k}, \quad 1 \le k \le n+m$$

then $N_{r+1}^\beta$ is activated and generates signals with Eq. (10).

### 3.2.3 Multimodal Association Neuron

When new input dimensions come from a new channel, multimodal association neurons absorb the new input dimensions, as the circled number 3 in Fig. 2 shows. In this situation, input from at least one original channel is needed to guide the input of the new channel to some proper multimodal association neurons.

Without loss of generality, we use an original channel $A$ which receives $\bar{\boldsymbol{x}}$ and a new channel $X$ which receives $\tilde{\boldsymbol{x}}$ to introduce the integration algorithm. As shown in Fig. 2, when each channel receives its input, it first conducts ascending activation using Eq. (1) and Eq. (3). Assume that feature neurons $N_{a_1}^{\alpha_1}, N_{a_2}^{\alpha_2}, ..., N_{a_n}^{\alpha_n}$ and unimodal association neuron $N_a^A$ are activated in channel $A$, feature neurons $N_{a_{n+1}}^{\alpha_{n+1}}, N_{a_{n+2}}^{\alpha_{n+2}}, ..., N_{a_{n+m}}^{\alpha_{n+m}}$ and unimodal association neuron $N_a^X$ are activated in channel $X$. The activated unimodal association neurons try to activate some multimodal association neuron with Eq. (5),

$$N_{\bar{a}} = \{N_i | \boldsymbol{W}_{i,:}^A \cdot \boldsymbol{e}^A = 1,\ N_i \in N\}, \quad N_a = \{N_i | \boldsymbol{W}_{i,:}^X \cdot \boldsymbol{e}^X = 1,\ N_i \in N\}$$

where $\boldsymbol{e}^A$ is a 0-1 vector and $\boldsymbol{e}_a^A = 1$ which means $N_a^A$ is activated, $\boldsymbol{e}^X$ is also a 0-1 vector and $\boldsymbol{e}_a^X = 1$ which means $N_a^X$ is activated. The integration algorithm finds a proper multimodal association neuron to connect the activated unimodal association neuron in channel $X$ according to the results of $N_{\bar{a}}$ and $N_a$. Again, there are four combinations:

(1) $N_{\bar{a}} \neq \varnothing$ and $N_a = \varnothing$, channel $A$ activates some multimodal association neuron, but the new channel $X$ does not. The network applies the dimension integration to $N_{\bar{a}}$ to associate the unimodal association neuron $N_a^X$ in channel $X$,

$$\boldsymbol{W}_{\bar{a},a}^X = 1 \tag{12}$$

$$\boldsymbol{U}_{\bar{a},a}^X = 1 \tag{13}$$

Eq. (12) expands the input dimension of the multimodal association neuron $N_{\bar{a}}$ by creating an ascending connection from the unimodal association neuron $N_a^X$ in the new channel $X$ to $N_{\bar{a}}$. Eq. (13) expands the output dimension of the multimodal association neuron $N_{\bar{a}}$ by creating a descending connection from $N_{\bar{a}}$ to $N_a^X$. Via the descending connection, the network can do retrieval among different channels. Together, Eq. (12) and Eq. (13) create a bidirectional connection between channel $A$ and the new channel $X$.

Then $N_{\bar{a}}$ can generate signals using the input from the new channel with Eq. (5),

$$[\boldsymbol{a}, \boldsymbol{\lambda}] = \mathcal{F}(\boldsymbol{z}^X) \tag{14}$$

Meanwhile, the descending connection $\boldsymbol{U}_{\bar{a},a}^A$ between $N_{\bar{a}}$ and $N_a^A$ adds $\boldsymbol{\lambda}$ of channel $X$ in Eq. (14) for transmitting signals from channel $X$, so does the descending connection $\boldsymbol{U}_{a,a_k}^{\alpha_k}$ between $N_a^A$ and the activated feature neuron $N_{a_k}^{\alpha_k}$. The ascending activation function of neuron $N_{\bar{a}}$ evolves from $n$ modalities to $n+1$ modalities, i.e.,

$$\exists \boldsymbol{W}_{\bar{a},:}^\beta \cdot \boldsymbol{e}^\beta = 1, \ \beta \in \{A, B\} \ \rightsquigarrow \ \exists \boldsymbol{W}_{\bar{a},:}^\beta \cdot \boldsymbol{e}^\beta = 1, \ \beta \in \{A, B, X\} \tag{15}$$

(2) $N_{\bar{a}} = \varnothing$ and $N_a \neq \varnothing$, channel $X$ activates some multimodal association neuron, but channel $A$ does not. We do a dimension integration process which is similar to that of combination (1) to the multimodal association neuron $N_a$.

(3) $N_{\bar{a}} \neq \varnothing$ and $N_a \neq \varnothing$, both channel $A$ and channel $X$ activate some multimodal association neurons. We do a dimension integration process similar to that of combination (1) to the multimodal association neurons $N_{\bar{a}}$ and $N_a$.

(4) $N_{\bar{a}} = \varnothing$ and $N_a = \varnothing$, neither channel $A$ nor channel $X$ activates any multimodal association neuron. A new association neuron $N_{r+1}$ is initialized to associate the unimodal association neurons $N_a^A$ and $N_a^X$ by creating new ascending and descending connections as follows,

$$\boldsymbol{W}_{r+1,a}^A = 1, \quad \boldsymbol{W}_{r+1,a}^X = 1, \quad \boldsymbol{U}_{r+1,a}^A = 1, \quad \boldsymbol{U}_{r+1,a}^X = 1 \tag{16}$$

Now the dimension integration for different types of neurons is introduced. It is easy to handle these different types of dimension integration simultaneously if the input contains different types of input dimension expansion, which will be verified in the experiments.

## 4 EXPERIMENTS

We use the datasets used in Xing et al. (2021) and Lai et al. (2011). The dataset Xing et al. (2021) contains images, audio (name words) and taste data of common fruits. We denote this dataset as **IAT** dataset. To verify the learning ability of the network, we add color words and taste words (audio) to **IAT**, resulting in an expanded dataset, named **eIAT**. The dataset Lai et al. (2011) contains images of objects in home environments. We take images of fruit objects from it and pair them with audio and taste data from **eIAT**. We denote this dataset as **MIX**. We compare classic and SOTA methods including PEN Xing et al. (2016), OPID Hou & Zhou (2018), AEN Xing et al. (2021), ALIU Tu et al. (2025), and OLD³S Lian et al. (2024). The parameters of our method are set as follows: In Eq. (1), $\theta$ of the feature neuron is set to a quarter of the 2-norm of the weight of the neuron and $T$ is set to 150. In Eq. (2) and Eq. (4), $\vartheta$ is set to 0.8 which means a relative probability of 80%. Visual features include the color of the object and the normalized Fourier descriptor of the object's shape. Auditory features are the Mel-Frequency Cepstral Coefficients of the syllables contained in the word.

**Channel Expansion:** Xing et al. (2021) design a channel expansion experiment. They first give their network a visual channel and let the network learn the visual samples. After a period of learning, an

Table 1: Results of the channel expansion experiment. V: Vision, A: Audition, T: Taste.

| Dataset | Task | Methods | | | | | |
|---|---|---|---|---|---|---|---|
| | | PEN | OPID | AEN | OLD$^3$S | ALIU | IDEN |
| **IAT** | V | $81.5 \pm 1.9$ | $84.0 \pm 2.1$ | $84.4 \pm 2.4$ | $86.3 \pm 2.1$ | $85.2 \pm 2.0$ | $\mathbf{88.9 \pm 1.9}$ |
| | A | $80.7 \pm 2.0$ | $81.5 \pm 2.0$ | $82.6 \pm 2.3$ | $84.1 \pm 1.9$ | $82.9 \pm 2.3$ | $\mathbf{86.1 \pm 2.2}$ |
| | T | $84.1 \pm 1.6$ | $88.0 \pm 1.7$ | $88.9 \pm 1.7$ | $89.5 \pm 1.6$ | $88.4 \pm 1.8$ | $\mathbf{91.8 \pm 1.5}$ |
| **eIAT** | V | $80.1 \pm 2.1$ | $80.9 \pm 2.4$ | $82.7 \pm 2.2$ | $83.9 \pm 1.8$ | $82.3 \pm 2.2$ | $\mathbf{86.0 \pm 1.9}$ |
| | A | $78.2 \pm 2.4$ | $78.7 \pm 2.2$ | $80.4 \pm 2.6$ | $81.2 \pm 2.0$ | $79.6 \pm 2.4$ | $\mathbf{84.7 \pm 2.1}$ |
| | T | $82.4 \pm 1.9$ | $85.2 \pm 1.6$ | $85.5 \pm 1.5$ | $86.3 \pm 1.3$ | $85.7 \pm 1.5$ | $\mathbf{88.4 \pm 1.7}$ |
| **MIX** | V | $72.9 \pm 2.6$ | $74.1 \pm 2.3$ | $75.5 \pm 2.5$ | $77.8 \pm 1.9$ | $75.0 \pm 2.3$ | $\mathbf{80.3 \pm 2.1}$ |
| | A | $70.3 \pm 3.1$ | $72.4 \pm 2.0$ | $74.3 \pm 2.9$ | $75.7 \pm 2.4$ | $73.9 \pm 2.4$ | $\mathbf{79.1 \pm 2.0}$ |
| | T | $74.4 \pm 2.5$ | $75.7 \pm 1.8$ | $76.8 \pm 2.3$ | $78.6 \pm 1.6$ | $76.4 \pm 2.1$ | $\mathbf{81.6 \pm 1.8}$ |

auditory channel is added to the network and pairs of object's image and audio sample are used to train the network. After all audio samples are learned, they add a taste channel to the network and use pairs of image and taste data to train the network. Here, to test the robustness of the methods, we do the channel expansion in random orders, e.g., we expand the channels in the order of taste, vision, and audition or audition, vision and taste, etc.

To test the learned model, we do a recall experiment, i.e., we feed input to one channel to get outputs from the other channels. For example, we use visual input to get auditory and taste outputs, which is denoted as **Task V** in Table 1. **Task A** means using auditory input to get visual and taste outputs, and **Task T** is defined similarly. OPID, ALIU, and OLD$^3$S can only perform classification after learning new dimensions, therefore, we conduct classification on these methods using the new dimension data. We conduct the training and testing experiments 30 times to obtain the average performance. Table 1 shows that our method gets the highest accuracy.

**Learning in Open Environment:** The above experiment is conducted in the close environment, where all classes are available beforehand. Here, to test the robustness of the methods against the **catastrophic forgetting**, we conduct the channel expansion in an open environment, where new classes emerge after the training is completed on the current classes. We divide the dataset into 2, 4, and 10 roughly equal parts with different classes. In each channel expansion step, we first feed one part to the network. After learning is completed, we feed the next part, and so forth. We also conduct the training and testing 30 times. Since OPID and OLD$^3$S require all classes available before training, they are not applicable to this environment. Table 2 shows that our method gets the highest accuracy, and is comparable to that in the closed environment shown in Table 1, indicating that our method is stable in the open environment.

**Mix Expansion:** The channel expansion in Xing et al. (2021) corresponds to the task of the dimension integration of the multimodal association neuron in our paper. Here, a mix expansion experiment is designed which is more general than channel expansion. It simultaneously uses the dimension integration of the feature neurons, unimodal association neurons and multimodal association neurons.

Table 2: Results in the open environment.

| Dataset | Parts | Task | Methods | | |
|---|---|---|---|---|---|
| | | | PEN | AEN | IDEN |
| **IAT** | 2 | V | $81.5 \pm 2.2$ | $84.3 \pm 2.0$ | $\mathbf{89.0 \pm 2.0}$ |
| | | A | $80.9 \pm 2.3$ | $82.8 \pm 2.5$ | $\mathbf{85.9 \pm 1.8}$ |
| | | T | $84.4 \pm 1.8$ | $89.1 \pm 1.8$ | $\mathbf{91.7 \pm 1.6}$ |
| | 4 | V | $81.7 \pm 2.0$ | $84.5 \pm 2.3$ | $\mathbf{88.7 \pm 1.9}$ |
| | | A | $81.2 \pm 2.5$ | $82.6 \pm 2.4$ | $\mathbf{86.2 \pm 2.0}$ |
| | | T | $84.3 \pm 1.6$ | $89.4 \pm 1.7$ | $\mathbf{91.7 \pm 1.7}$ |
| | 10 | V | $82.1 \pm 1.8$ | $84.8 \pm 2.1$ | $\mathbf{89.3 \pm 2.0}$ |
| | | A | $81.6 \pm 2.1$ | $83.2 \pm 2.4$ | $\mathbf{86.4 \pm 1.9}$ |
| | | T | $84.8 \pm 1.7$ | $89.8 \pm 1.6$ | $\mathbf{92.3 \pm 1.6}$ |
| **eIAT** | 2 | V | $80.4 \pm 2.0$ | $82.9 \pm 2.1$ | $\mathbf{86.3 \pm 2.1}$ |
| | | A | $78.1 \pm 2.4$ | $80.8 \pm 2.6$ | $\mathbf{84.5 \pm 2.2}$ |
| | | T | $82.2 \pm 2.3$ | $85.7 \pm 1.7$ | $\mathbf{88.7 \pm 1.7}$ |
| | 4 | V | $80.5 \pm 2.1$ | $83.1 \pm 2.3$ | $\mathbf{86.4 \pm 1.9}$ |
| | | A | $78.2 \pm 2.5$ | $81.0 \pm 2.4$ | $\mathbf{84.8 \pm 2.0}$ |
| | | T | $82.5 \pm 1.9$ | $85.4 \pm 1.8$ | $\mathbf{88.5 \pm 1.6}$ |
| | 10 | V | $80.9 \pm 1.9$ | $83.5 \pm 2.2$ | $\mathbf{86.6 \pm 2.0}$ |
| | | A | $78.8 \pm 2.5$ | $81.4 \pm 2.5$ | $\mathbf{85.2 \pm 2.2}$ |
| | | T | $83.2 \pm 2.3$ | $86.1 \pm 1.6$ | $\mathbf{89.1 \pm 1.7}$ |
| **MIX** | 2 | V | $72.8 \pm 3.3$ | $75.8 \pm 2.8$ | $\mathbf{80.5 \pm 2.3}$ |
| | | A | $70.6 \pm 3.1$ | $74.1 \pm 3.1$ | $\mathbf{79.0 \pm 2.3}$ |
| | | T | $74.7 \pm 2.5$ | $76.9 \pm 2.0$ | $\mathbf{81.4 \pm 1.9}$ |
| | 4 | V | $73.0 \pm 3.4$ | $76.1 \pm 2.6$ | $\mathbf{80.7 \pm 2.2}$ |
| | | A | $70.5 \pm 3.2$ | $74.4 \pm 2.8$ | $\mathbf{79.2 \pm 2.1}$ |
| | | T | $74.9 \pm 2.7$ | $77.5 \pm 2.0$ | $\mathbf{81.6 \pm 2.0}$ |
| | 10 | V | $73.5 \pm 3.2$ | $76.6 \pm 2.5$ | $\mathbf{81.1 \pm 2.2}$ |
| | | A | $71.1 \pm 3.1$ | $75.0 \pm 2.7$ | $\mathbf{79.7 \pm 2.3}$ |
| | | T | $75.2 \pm 2.6$ | $77.8 \pm 2.1$ | $\mathbf{82.2 \pm 1.8}$ |

For example, we initially give our network a visual channel that only receives the gray image of each object. At this stage, we let the network learn shape features of the object. After a period of learning, we add green and blue dimensions to the visual channel and the network can receive

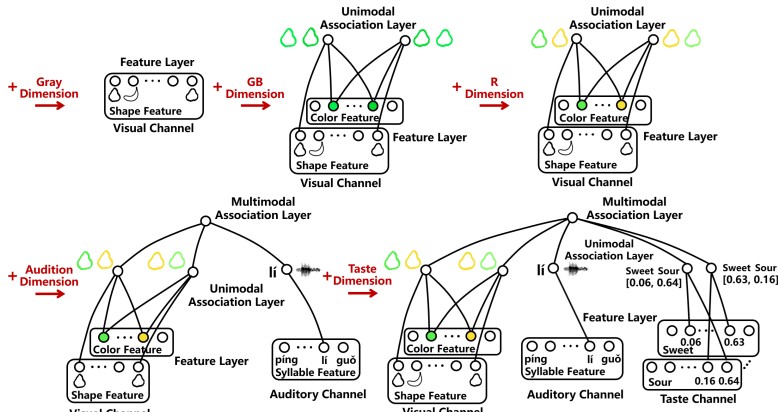

Figure 3: Change of a substructure of the network after new input dimensions are added. The icons next to the neurons (circles) represent the objects to which the neurons maximally respond.

gray and GB images. We let the network learn color features from the GB images. Meanwhile, the network should correctly introduce the color features to the learned shape features. Next we add a red dimension to the visual channel. The network should correctly integrate the red dimension

color features to the learned GB dimension color features. After that, we add an auditory channel to the network. In this period, pairs of audio sample and image of each object are fed into the network to integrate the auditory channel. Finally, a taste channel is added to the network. We also use different adding orders of above dimensions and do the experiment 30 times in close and open environments. All comparison methods cannot handle this mix expansion experiment, we do not train them. Table 3 shows that the accuracy of the mix expansion is comparable with that in the channel expansion in both environments, which is higher than the accuracy of all other methods in Table 1 and Table 2.

Table 3: Results of the mix expansion.

| Dataset | Env. | Task | | |
|---|---|---|---|---|
| | | V | A | T |
| IAT | Close | 88.5 ± 2.0 | 85.8 ± 1.9 | 91.4 ± 1.8 |
| | Open | 89.0 ± 2.1 | 86.1 ± 2.0 | 92.1 ± 1.7 |
| eIAT | Close | 85.8 ± 2.2 | 84.0 ± 2.2 | 88.0 ± 1.6 |
| | Open | 86.3 ± 2.0 | 84.9 ± 2.2 | 88.7 ± 1.8 |
| MIX | Close | 80.1 ± 2.5 | 78.7 ± 2.3 | 81.1 ± 2.1 |
| | Open | 80.8 ± 2.3 | 79.4 ± 2.5 | 81.9 ± 1.9 |

Fig. 3 shows the change of a substructure of the network with the expansion of the input dimensions. In the beginning the network learns some shape features for the object pear. After the GB dimension is added, the network learns two color features in the green-blue color space. Two visual unimodal association neurons are obtained by integrating the color features with the shape features. After the red dimension is added, the two color feature neurons are expanded to the RGB color space. The two visual unimodal association neurons are also expanded to RGB dimension which represent a green pear and a yellow pear. After the auditory channel is added, the neurons in the visual channel are associated with the auditory channel. The network comes to a new world with the concept of auditory sense. Finally, the taste channel is correctly added to the network. Fig. 3 demonstrates that new dimensions are correctly and effectively integrated into the network in an online way.

## 5 CONCLUSION

To enable multi-level dimension expansion, a hierarchical modular network inspired by the brain network structure is designed, and an integration algorithm is proposed to bind the new input dimensions with the existing ones in an online manner. We design a series of comprehensive dimension expansion experiments, and the results confirm our claims. Our method is capable of handling many potential practical problems in fields such as robotic systems, information fusion, and data stream mining. For example, when new sensors are added to a robot to expand its perception capabilities, there is no need for tedious offline retraining of the robot from scratch. Instead, the information collected by the new sensors will be automatically absorbed by the network in an online manner.

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
