# OpenReview forum: "Input Dimension Expandable Network: Integrating New Input Dimensions in Online Learning"
_ICLR.cc/2026/Conference — ICLR 2026 Conference Withdrawn Submission_

### Official Review · Reviewer_Dm66 · 2025-10-27

**Soundness:** 2
**Presentation:** 2
**Contribution:** 2
**Rating:** 2
**Confidence:** 3

**Summary:**

This paper introduces the Input Dimension Expandable Network (IDEN), a novel hierarchical and modular neural network designed to handle the expansion of input dimensions during online learning without requiring complete retraining. Inspired by the perceptual plasticity of the mammalian brain, the proposed architecture can integrate new information at multiple levels: adding new features to an existing input type, adding new feature types to a modality, or adding entirely new modalities. The authors propose a dimension integration algorithm that guides how new inputs are incorporated into the existing network structure. The method's effectiveness is demonstrated through a series of experiments on recombiniations of a multi-modal dataset, showing superior performance over existing methods in tasks involving channel expansion, online classification, and mixed-modality expansion.

**Strengths:**

This work seems to present some novel ideas towards continual learning, enabling expansion of input dimensionalities and modalities. In the experimentational results presented, it outperforms existing related methods.

**Weaknesses:**

Although three datasets are presented using combinations of data from two sources, they overlap in their samples and features, and the "tasks" seem to only be classification. This fails to demonstrate the method's generalizability and robustness.

The results tables are presented without defining the metric being measured (some sort of accuracy?), nor clearly specifying the task being performed and whether these results were run on holdout test splits of the dataset. These errors make the empirical results difficult to properly interpret or trust. Evaluating the models on holdout data is crucial to show the generalization abilities of the implemented methods: my understanding of your methods is like a nearest-neighbor lookup, but such methods are prone to overfitting on training data without generalizing to unseen examples.

Minor edits:
* The citation entitled "Learning with feature evolvable stream" is repeated twice with different publication years.
* Use `\citep` whenever the citation is not directly part of the sentence structure, for example "...through brain computer interface (Thomson et al., 2017) and genetic engineering (Zhang et al., 2017)...".

**Questions:**

How does your proposed model actually "learn" the weights outside of the architectural growth? Or are these generated randomly? In this case, how does the model control the computational complexity of expansion?

What are the implementational details of your methods? In the current state of the paper, understanding the computational scale, let alone trying to use these methods and/or replicate the results, is difficult without many more details, such as initial and final architectural dimensions as well as dataset sizes. Only a few paremeter settings are given in the beginning of Section 4.

In the related works section, you state that other works are "unable to perform classification using data from the original dimensions after new dimensions have been introduced", but where are the results showing that your method can do so? Your results tables seem to only show some kind of overall accuracy, not splitting out (for example) performance on each partition of classes in Table 2.

---

### Official Review · Reviewer_WnAf · 2025-10-30

**Soundness:** 2
**Presentation:** 2
**Contribution:** 2
**Rating:** 4
**Confidence:** 3

**Summary:**

This paper proposes a hierarchical modular neural network capable of expanding input dimensions in an online manner.  It is a multi-channel structure, where each channel consists of several feature areas and a unimodal association layer. Then all channels are  integrated using a multimodal association layer. An online integration algorithm is designed to bind new input dimensions with existing representations dynamically.  Corresponding experiments are designed  to realize and verify the dimension expandable learning paradigm.

**Strengths:**

1.The paper addresses a fundamental yet underexplored problem: how to enable a learning system to expand its input dimensions in an
online manner? The connection to biological inspiration is appealing.
2.The proposed mechanism has strong potential in real-world applications such as adaptive robots or lifelong learning systems where new modalities or sensors are continuously introduced.
3.The dimension expansion experiments demonstrate that the proposed framework can integrate new dimensions in an online manner, which is an important step toward adaptive systems.

**Weaknesses:**

1.The paper spends substantial space describing the model architecture and algorithmic procedures, yet provides limited theoretical analysis and lacks clear explanation for the underlying design motivations.
2.The “brain-inspired” aspect is mentioned but not rigorously analyzed. It would be stronger if specific neuroscientific parallels were empirically or theoretically grounded.
3.The experiments are limited and do not include ablation studies. As a result, the empirical section does not convincingly demonstrate the effectiveness of each proposed component or justify the necessity of the hierarchical modular design.
4.Figure 2 annotation errors and unclear labeling.

**Questions:**

1.How is representational stability guaranteed when integrating new dimensions?
2.What is the computational overhead of frequent dimension expansions?
3.Please correct and unify the annotations in Figure 2.

---

### Official Review · Reviewer_e1Nj · 2025-10-31

**Soundness:** 3
**Presentation:** 3
**Contribution:** 3
**Rating:** 4
**Confidence:** 4

**Summary:**

The proposed Input Dimension Expandable Network (IDEN( is a hierarchical, modular neural architecture expanding its input space online at three levels: (i) adding new dimensions to existing features, (ii) adding new feature types within a modality (unimodal association), and (iii) adding entirely new modalities (multimodal association). A dimension integration algorithm routes and binds new inputs to existing circuits via ascending/descending pathways, with frequency-tagged signaling and Fourier-based aggregation at the multimodal layer. Experiments on small multimodal datasets evaluate channel expansion, open-environment learning, and a mix expansion setting. Results on limited datasets outperform selected baselines.

**Strengths:**

IDEN is a brain-inspired modular hierarchy model mimicing the organization of the mammalian sensory system and supporting online “plug-and-play”. It supports feature-level, unimodal, and multimodal expansion, a novel innovation not seen in many prior works.

It tackles open feature space learning with multi-level and multi-region expansion in a single unified framework. It addresses limitations in existing models by providing a unified framework for three distinct types of input expansion: adding new features to an existing input, adding new feature types within a modality, and adding entirely new modalities.

The dimension integration algorithm  how existing circuits absorb new inputs without full retraining, supporting both upward (perception) and downward (feedback/recall) flows. This allows adaptation to new input types during operation, enabling continual growth.

**Weaknesses:**

The framework lacks theoretical guarantees with no analysis or proof of convergence, stability during dimension growth, prevention of catastrophic forgetting.

The datasets tested on are small and synthetic, with simple sensory modalities, so It’s not clear whether the model would scale to real-world multimodal or high-dimensional data

The paper overlooks computational cost and scaling results that rae very crucial for an online model. The experiments demonstrate performance but do not analyse how internal representations evolve when new dimensions are added.

While inspired by neuroscience, the connection is more metaphorical than mechanistic. It has heavy mathematical notations (cosine activations, Fourier encoding, Gaussian likelihood gates) making the method hard to reimplement. The maths is dense and symbol-heavy, with minimal intuitive explanation.

The paper has long paragraphs, repetitive sections, and also lacks pseudocode. They also lacks visualisation and  quantitative metrics for representational consistency.

**Questions:**

How does the dimension integration algorithm handle noisy or contradictory information from a new input source? For example, if a newly added sensor provides data that conflicts with well-established patterns from existing sensors, is there a mechanism to resolve this conflict or weight the more reliable source?

The experiments are focused on classification and cross-modal recall tasks. How do you envision the IDEN architecture being applied to other machine learning paradigms, such as reinforcement learning, where an agent would need to learn to associate new sensory inputs with optimal actions and reward signals?

Can the authors quantify computational overhead as the network expands?

Can the authors provide a visualisation and  pseudocode describing the algorithm?

---

### Official Review · Reviewer_Jtsv · 2025-11-04

**Soundness:** 1
**Presentation:** 1
**Contribution:** 3
**Rating:** 0
**Confidence:** 3

**Summary:**

This article describes a neural network architecture meant to support online learning of new dimension in its input. For this, a dedicated neural network architecture

**Strengths:**

The paper addresses an interesting problem with an unusual neural network architecture, namely, the addition and online adaptation to  new input dimensions. The experiments support that the claims of the paper using a dataset that seems to be constructed from multiple high-dimensional modalities (although these are preprocessed further).

**Weaknesses:**

- The motivation for input expansion is quite weak: enhancing animals' perception seems to work via existing cells, although they react to new stimuli
- the paper is not very clearly written. Especially the description of the proposed network is quite unclear. Is this a recurrent network (since there is a "period time")? How are ascending and descending pathways integrated?
- many experimental details remain quite vague: nature of the data, the features that are fed to the network, number of data samples, training method, parameters justification, parameters of baselines, ...
- it is not clear how this network architecture would perform, e.g., classification tasks

**Questions:**

- Could you please give a link where the benchmark datasets can be obtained?
- what would an application in the real world look like that profits from this kind of technique?
- please give more information about the used datasets: dimensionality, number of samples, etc...
- what are features that are actually received by model neurons? how many, what is their nature?
- please explain how such a network is trained, I found no details in the paper. Training algorithm (eg, SGD), optimizer(Adam?), learning rates, learning rate schedule, ...
- what are parameters for the baseline methods? How did you tune them?
- how did you arrive at the parameters given in Sec.4? Cross-validation? On what data?
- how would this network perform, e.g., classification?
- can this network also operate on raw data?

---

### Note · Authors · 2025-11-13

I have read and agree with the venue's withdrawal policy on behalf of myself and my co-authors.